# ID1-Mediated BMP Signaling Pathway Potentiates Glucagon-Like Peptide-1 Secretion in Response to Nutrient Replenishment

**DOI:** 10.3390/ijms21113824

**Published:** 2020-05-28

**Authors:** Jae Woong Jeong, Minki Kim, Jiwoo Lee, Hae-Kyung Lee, Younhee Ko, Hyunkyung Kim, Sungsoon Fang

**Affiliations:** 1Department of Medicine, Yonsei University College of Medicine, Seoul 03722, Korea; jjown@yonsei.ac.kr; 2Department of Medical Science, BK21 Plus Project for Medical Science, Yonsei University College of Medicine, Seoul 03722, Korea; louiskim100@yonsei.ac.kr; 3Severance Biomedical Science Institute, Gangnam Severance Hospital, Yonsei University College of Medicine, Seoul 03722, Korea; hylee609@gmail.com (J.L.); hklee@yuhs.ac (H.-K.L.); 4Division of Biomedical Engineering, Hankuk University of Foreign Studies, Yongin 17035, Korea; younko@hufs.ac.kr; 5Department of Biochemistry and Molecular Biology, Korea University College of Medicine, Seoul 02841, Korea

**Keywords:** glucagon-like peptide-1, L cells, bone morphogenetic protein 4, incretin, inhibitor of DNA binding 1

## Abstract

Glucagon-like peptide-1 (GLP-1) is a well-known incretin hormone secreted from enteroendocrinal L cells in response to nutrients, such as glucose and dietary fat, and controls glycemic homeostasis. However, the detailed intracellular mechanisms of how L cells control GLP-1 secretion in response to nutrients still remain unclear. Here, we report that bone morphogenetic protein (BMP) signaling pathway plays a pivotal role to control GLP-1 secretion in response to nutrient replenishment in well-established mouse enteroendocrinal L cells (GLUTag cells). Nutrient starvation dramatically reduced cellular respiration and GLP-1 secretion in GLUTag cells. Transcriptome analysis revealed that nutrient starvation remarkably reduced gene expressions involved in BMP signaling pathway, whereas nutrient replenishment rescued BMP signaling to potentiate GLP-1 secretion. Transient knockdown of inhibitor of DNA binding (ID)1, a well-known target gene of BMP signaling, remarkably reduced GLP-1 secretion. Consistently, LDN193189, an inhibitor of BMP signaling, markedly reduced GLP-1 secretion in L cells. In contrast, BMP4 treatment activated BMP signaling pathway and potentiated GLP-1 secretion in response to nutrient replenishment. Altogether, we demonstrated that BMP signaling pathway is a novel molecular mechanism to control GLP-1 secretion in response to cellular nutrient status. Selective activation of BMP signaling would be a potent therapeutic strategy to stimulate GLP-1 secretion in order to restore glycemic homeostasis.

## 1. Introduction

Obesity and type II diabetes (T2D) are the most common metabolic syndromes that are prevalent worldwide. T2D patients generally exhibit symptoms of disturbed metabolic functions in various organs, such as hyperglycemia and insulin resistance [1,2,3,4]. Glucagon-like peptide-1 (GLP-1) is an incretin hormone that strongly potentiates insulin secretion, promotes the growth and survival of pancreatic β cells, and controls physiological glycemic homeostasis [5,6]. Given that GLP-1 controls glycemic controls, there are common views demonstrating that T2D patients often exhibit impaired GLP-1 secretion [7]. Therefore, GLP-1 therapies for T2D patients have been reported as well-established therapeutic strategies for the treatment of T2D patients [8,9,10].

Previous reports have demonstrated that GLP-1 secretion is modulated by various nutrients, such as glucose, glutamine, and protein ligand signals [11,12]. In the gastrointestinal tract, nutrient replenishment followed by food intake stimulates bile acid secretion from the gall bladder to promote fat digestion, lipid emulsification, and lipid absorption. Consistently, it has been demonstrated that bile acids activated bile acid receptors, including farnesoid X receptor (FXR) and G protein-coupled bile acid receptor 1 (GPBAR1; also known as TGR5), to potentiate GLP-1 secretion in L cells, suggesting that bile acid signaling pathway would be a plausible molecular mechanism to control GLP-1 secretion in response to nutrient status in L cells [13,14,15]. However, the detailed molecular mechanisms of how GLP-1 secretion is controlled in L cells in response to nutritional stress still remains unclear.

Previously, it has been reported that activation of bile acid signaling upregulates the expression level of bone morphogenetic protein 4 (BMP4) [16]. BMPs, members of the transforming growth factor β (TGF-β) superfamily, are multi-functional growth factors that play pivotal roles in bone formation, embryogenesis, and development [17,18]. BMP signaling pathway induces activation of SMAD family members, such as SMAD1, SMAD5, and SMAD9 (called as SMAD8), and controls gene expression of inhibitor of DNA binding (ID). Recently, it has been reported that BMP4 plays a key role to control metabolic homeostasis to mediate white adipose tissue (WAT) browning, and prevents obesity and insulin resistance in animal models [19,20]. These data clearly suggest that BMP4 might have beneficial impacts to control physiological metabolic homeostasis, such as glycemic homeostasis. Surprisingly, it has been also reported that T2D patients exhibit relatively higher levels of circulating BMP4 [20,21]. Furthermore, it has been shown that obesity is positively correlated with cellular BMP4 resistance, implying that BMP4 may participate in signaling pathways to control metabolic homeostasis [19]. Although a previous report has demonstrated that treatment of GLP-1 agonist decreased the level of BMP4 [22], physiological effects of BMP4 signaling to control GLP-1 secretion in L cells is still unclear.

In this paper, we found that activation of the BMP signaling pathway is able to potentiate GLP-1 secretion in response to nutrient replenishment. We investigated the molecular mechanisms of GLP-1 secretion according to three different nutrient status (normal, starvation, and replenishment) using GLUTag, a model of murine enteroendocrinal L cells. RNA transcriptome analysis revealed that ID1, the target gene of BMP signaling, was significantly downregulated in nutrient starvation. Nutritional replenishment, however, upregulated ID1 gene expression. Quite interestingly, we observed that the regulation of ID1 gene expression was similar with the pattern of GLP-1 secretion in response to nutritional stresses, implying that ID1-mediated BMP signaling would be critical to control GLP-1 secretion in response to nutritional status in L cells. Consistently, inhibition of BMP signaling pathway using LDN193189, a BMP receptor inhibitor, exhibited reduction of ID1 gene expression and GLP-1 secretion. Moreover, activation of BMP signaling pathway with BMP4 markedly potentiated GLP-1 secretion in response to nutritional replenishment. Altogether, we demonstrated that BMP signaling pathway is a novel molecular mechanism to control GLP-1 secretion in response to cellular nutrient status. We propose that selective activation of BMP signaling pathway in L cells would be a novel therapeutic strategy to potentiate GLP-1 secretion, leading to restoration of glycemic homeostasis in T2D patients.

## 2. Results

### 2.1. Nutritional Stress Rapidly Modulated GLP-1 Secretion and Cellular Bioenergetics in GLUTag Cells

To determine intracellular mechanisms to control GLP-1 secretion in response to nutrient status, we cultured enteroendocrinal L cells, GLUTag cells with three different nutrient statuses: (1) normal condition, (2) nutrient starvation, and (3) nutrient replenishment (Figure 1A). Although GLP-1 secretion was remarkably decreased in the condition of nutrient starvation, GLP-1 secretion was significantly increased in GLUTag cells by nutrient replenishment (Figure 1B). Given that cellular respiration is highly correlated with GLP-1 secretion [23,24,25], we next measured cellular bioenergetics (Figure 1C). We first measured basal cellular energy metabolism and found that the basal respiration was remarkably reduced by starvation for 2 h (Figure 1D). Subsequent nutrient replenishment after starvation remarkably rescued cellular respiration (Figure 1D). In the next step, we examined the capacity of the cells to modulate their bioenergetics in response to pharmacological metabolic modulators with nutritional stress. We sequentially treated the inhibitor of the mitochondrial ATP synthase oligomycin; the ionophore FCCP; and the inhibitors of the respiratory chain complexes III and I, antimycin and rotenone [26]. Consistently, maximal respiration capacity as well as reserve capacity in nutrient starvation remarkably reduced in GLUTag cells compared to those of normal or nutrient replenishment (Figure 1E). Furthermore, cells with nutrient starvation potently reduced ATP synthase-coupled respiration in response to oligomycin compared to those of normal or nutrient replenishment (Figure 1F). These data clearly suggest that nutritional stress rapidly controls GLP-1 secretion and cellular bioenergetics in GLUTag cells.

### 2.2. Nutrient Starvation Repressed BMP Signaling Pathway in GLUTag Cells

Next, we harvested RNAs from GLUTag cells with different nutrient statuses for further transcriptome analysis. To interrogate the transcriptional signatures affected by nutrient conditions of normal, starvation, and replenishment, we applied clustering for genes on the basis of complete-linkage clustering and Pearson’s correlation. Although gene expression patterns of normal condition were generally similar to those of nutrient replenishment, gene expression profiles of nutrient starvation were quite distinct from those of normal condition and nutrient replenishment (Figure 2A), implying that nutrient starvation for 2 h was a profound nutritional stress to disturb cellular metabolic homeostasis. We next performed a volcano plot analysis to identify changes of differentially expressed genes (DEG) from nutrient starvation. Regarding cut-off value for gene expression fold change (FC) of 2, we hypothesized that genes with a FC value less than −2 were downregulated, whereas genes with a FC value more than 2 were upregulated. The volcano plot analysis revealed that gene expression patterns were distinctly separated by nutrient starvation (Figure 2B). We also observed that 96 genes were upregulated, whereas 102 genes were significantly downregulated in the condition of nutrient starvation compared to those of normal/nutrient replenishment conditions (Figure 2C). KEGG pathway analysis revealed that signaling pathways, such as transforming growth factor-β (TGF-β) and mitogen-activated protein kinase (MAPK) signaling pathways were remarkably changed in the condition of nutrient starvation in GLUTag cells (Figure 2D). Given that bone morphogenetic protein (BMP) is a unique extracellular multifunctional signaling molecule belonging to the TGF-β superfamily [27], gene set enrichment analysis (GSEA) revealed that expression of genes involved in regulation of BMP signaling, including BMP receptor 1A (BMPR1A) and ID1, were significantly enriched (Figure 2E). Consistent with GSEA, a heatmap of core enriched gene expression revealed that gene expressions involved in BMP signaling pathways were remarkably downregulated in nutrient starvation compared to those of normal/nutrient replenishment conditions, implying that BMP signaling pathway would be involved in modulation of cellular homeostasis in response to nutritional stress (Figure 2E). Next, we performed functional protein association network analysis using STRING, with the cut-off value for combined score being 0.4. Consistent with GSEA analysis, numerous genes involved in BMP signaling pathway were identified as contributing to the protein functional network. Although downregulated in nutrient starvation, NOG, ID1, ID2, ID3, ID4 and SMAD6 genes were hub genes to control BMP signaling pathway in response to nutrient status (Figure 2F). Altogether, our data clearly suggest that BMP signaling pathway is crucial for maintaining cellular metabolic homeostasis in response to nutritional stress.

### 2.3. ID1-Mediated BMP Signaling Modulated GLP-1 Secretion in L Cells

Given that NOG, ID1, ID2, ID4, and SMAD6 genes were identified as hub genes to control BMP signaling pathway in response to nutritional stress, we next analyzed DEG from normal condition and nutrient starvation to examine of how these genes were regulated in response to nutritional stress (Figure 3A). Consistently, we observed that expression of genes such as ID1, ID4, and NOG was markedly downregulated in nutrient starvation with volcano plot analysis and heatmap of gene expression (Figure 3A,B), implying that BMP signaling pathway is rapidly changed within 2 h nutritional stress. To determine how many genes are regulated by nutrient starvation, we drew a Venn diagram using DEGs. We noticed that 82 genes were overlapped DEGs between normal vs. starvation and starvation vs. replenishment (Appendix A). We next analyzed DEG from nutrient starvation and nutrient replenishment (Figure 3C). Although downregulated in nutrient starvation, gene expressions of ID1, ID2, ID3, ID4, SMAD6, SMAD7, and NOG were significantly upregulated by nutrient replenishment (Figure 3C,D). Consistently, we confirmed that gene expression of ID1, ID4, and NOG were remarkably downregulated in nutrient starvation, whereas nutrient replenishment rescued those gene expressions by real-time PCR among hub genes (Figure 3E). Although no changes were observed in the protein levels of ID4 (Appendix A), immunoblotting confirmed that the protein level of ID1 was decreased significantly in nutrient starvation (Figure 3F), implying that nutritional stress controlled BMP signaling pathway-related gene ID1 at the levels of transcription and translation in GLUTag cells. To determine the functional roles of ID1 gene for GLP-1 secretion, we first transiently knocked down ID1 gene using siRNA. We confirmed that siRNA against ID1 gene remarkably reduced the levels of both mRNA and protein (Figure 3G,H). Quite interestingly, we observed that transient knockdown of ID1 gene significantly reduced GLP-1 secretion and cellular bioenergetics (Figure 3I,J), suggesting that ID1 is crucial to control GLP-1 secretion in GLUTag cells.

### 2.4. Modulation of BMP Signaling Pathway Controlled GLP-1 Secretion in GLUTag Cells

To examine whether BMP signaling pathway directly controls GLP-1 secretion, we next treated a BMP signaling pathway inhibitor, LDN193189, to suppress BMP signaling pathway in L cells. As expected, LDN193189 treatment remarkably downregulated expression of ID1 genes (Figure 4A). Consistent with conditions of nutritional starvation, we also observed that LDN193189 treatment markedly reduced level of phosphorylated SMAD1/5/9 (Figure 4B). We next performed GLP-1 secretion in LDN193189-treated L cells. Although nutrient replenishment rescued secretion of GLP-1 in vehicle-treated GLUTag cells, inhibition of BMP signaling pathway by LDN193189 remarkably reduced GLP-1 secretion in response to nutrient replenishment (Figure 4C). These data clearly demonstrated that BMP signaling pathway is crucial to potentiate GLP-1 secretion in response to nutritional stress in GLUTag cells.

Finally, we examined whether activation of BMP signaling pathway may stimulate GLP-1 secretion in GLUTag cells. Although BMP4 treatment slightly reduced ID1 gene expression in starvation, we observed that BMP4 was able to upregulate expression of ID1 genes in GLUTag cells with nutrient replenishment (Figure 4D). Quite interestingly, we also observed that BMP4 treatment increased phosphorylated level of SMAD1/5/9 in nutrient starvation and nutrient replenishment, whereas no changes were observed in normal condition (Figure 4E), implying that BMP4 would be crucial to activate BMP signaling pathway in the conditions of nutrient starvation and replenishment. Consistent with phosphorylated SMAD1/5/9, protein level of ID1 was increased by BMP4 treatment in the condition of starvation and nutrient replenishment (Figure 4E). We next measured GLP-1 secretion in vehicle-treated and BMP4-treated GLUTag cells. Although no changes were observed in normal condition, we observed that BMP4 treatment slightly increased GLP-1 secretion in nutrient starvation (Figure 4F). Strikingly, BMP4 treatment significantly potentiated GLP-1 secretion in the condition of nutrient replenishment compared to that of vehicle treatment (Figure 4F). Altogether, we demonstrated that BMP signaling pathway is crucial to control GLP-1 secretion in GLUTag cells in response to nutritional stress.

## 3. Discussion

GLP-1 therapy is a very effective and well-established treatment for T2D and obesity in clinic [7]; however, the molecular mechanisms of how GLP-1 is secreted in physiological nutrient condition is unclear. With massive transcriptome analysis, we suggest that ID1-mediated BMP signaling pathway may play an important role to modulate GLP-1 secretion in response to nutritional status. Transient knock-down of ID1 gene exhibited reduction of GLP-1 secretion and cellular bioenergetics in GLUTag cells. Consistently, LDN193189-mediated inhibition of BMP signaling pathway reduced GLP-1 secretion in GLUTag cells. Interestingly, potentiation of BMP signaling pathway with BMP4 stimulated GLP-1 secretion under nutrient replenishment (Figure 5).

The physiological impact of ID1 gene in GLP-1 secretion has not been reported yet. Previously, it has been reported that ID1-deficient mice are resistant to diet-induced glucose intolerance and insulin resistance [28]. Additionally, transient knockdown of ID1 gene has been reported to potentiate insulin secretion in murine pancreatic β cells (MIN 6 cells) [28]. Thus, deficiency of ID1 gene increased insulin secretion to protect against hyperglycemia in an animal model. In our study, transient knockdown of ID1 gene seemingly reduced GLP-1 secretion in GLUTag cells. The discrepancy of ID1 gene functions between pancreatic β cell and enterodendocrinal GLUTag cells would have been the result of environmental difference, such as microbiota. It has been recently reported that the gut microbiome modulates bile acid composition to control bile acid signaling to potentiate GLP-1 secretion in L cells [14]. Given that GLUTag cells are derived from the murine intestinal tract, it is highly possible that the gut microbiome modified ID1-mediated secretory mechanisms in GLUTag cells to potentiate GLP-1 secretion in response to nutrient replenishment.

We observed that BMP treatment slightly reduced ID1 gene expression in GLUTag cells with 2 h starvation. However, the reduction of ID gene by BMP treatment was very marginal, suggesting that BMP4 treatment may not have “biological significance” on ID1 gene expression in GLUTag cells with 2 h nutritional starvation. In addition to ID1, we noticed that mRNA expression of ID4 was markedly changed by nutritional status. Although protein levels of ID4 were not changed by nutritional stress, we noticed that transient knockdown of ID4 was able to reduce ID4 mRNA level and GLP-1 secretion in GLUTag cells (Appendix A), suggesting that ID4 may partially participate in regulation of GLP-1 secretion in GLUTag cells. Likewise, we also noticed that mRNA expression of Nog was remarkably changed by nutritional status. Nog has been reported to serve as an inhibitor of BMP signaling pathway [29], implying that multiple factors involved in BMP signaling pathway would participate in the regulation of GLP-1 secretion in GLUTag cells in response to diverse nutritional status. Thus, it is plausible that inhibition of multiple genes involved in BMP signaling pathway would have synergistic effects to dramatically suppress GLP-1 secretion in GLUTag cells.

Previously, the correlation of GLP-1 and BMP-4 has been investigated [22]. It has been demonstrated that serum level of BMP-4 was positively correlated with obesity [22]. Quite interestingly, treatment of the GLP-1 receptor agonist exenatide markedly decreased serum level of BMP4 in obese patients. In our data, BMP4 was shown to potentiate GLP-1 secretion in L cells. Given that obese patients are under metabolic stress with reduction of GLP-1 secretion, it is highly plausible that BMP4 level would be elevated to increase GLP-1 secretion in L cells of obese patients.

Bile acid-activated TGR5 has been reported to stimulate cAMP signaling pathway to potentiate GLP-1 secretion in L cells [13,15]. A previous study has reported that BMP4-mediated signaling pathway is associated with cAMP signaling pathway [30]. It has also been reported that cAMP signaling pathway enhances SMAD-mediated BMP signaling pathway [31]. As previously reported, cAMP pathway is crucial to stimulate GLP-1 secretion mechanisms in response to glucose in L cells. Therefore, it is plausible that enhancement of cAMP signaling may potentiate BMP signaling pathway to potentiate GLP-1 secretion. It has also been reported that dysregulation of bile acid signaling became worse in patients with non-alcoholic fatty liver disease (NAFLD) and non-alcoholic steatohepatitis (NASH) [32]. Moreover, NAFLD and NASH patients have been reported to have deficiency in glucose-stimulated GLP-1 secretion [33]. Thus, it is possible that dysregulated bile acid signaling in NAFLD and NASH patients cannot activate TGR5 in L cells, leading to aggravate disrupted cAMP and BMP signaling for GLP-1 secretion in response to nutrients such as glucose.

## 4. Materials and Methods

### 4.1. Reagent

For activation of BMP signaling pathway, BMP4 peptide (SRP3298-10UG) was purchased from Sigma-Aldrich (St. Louis, MO, USA). For inhibition of BMP signaling, LDN193189 (SML0559-5MG) were purchased from Sigma-Aldrich. Sitagliptin (1757-100), a DPP4 inhibitor, was purchased from BIOVISION (1757-100).

### 4.2. Cell Line

GLUTag cell line (a well-known murine enteroendocrine L cell line) was kindly provided from Drucker lab. GLUTag were maintained in Dulbecco’s modified Eagle’s medium (DMEM; SH30243.01, Hyclone, LM 001-05, Welgene, Gyeongsan-si, Korea) with 10% fetal bovine serum (FBS; 35-015-CV, Corning at Corning, NY, USA) and 1% penicillin-streptomycin (PS; 15140122, Gibco at Waltham, MA, USA) at 37 °C at 95% humidified air and 5% CO_2_. Cells were passaged with 0.25% Trypsin-EDTA (25200072, gibco) and seeded on a 100 mm culture dish (353003, Falcon at Waltham, MA, USA). Nutrient-full media were based on DMEM with glucose (LM 001-05, Welgene), 10% FBS, and 1% PS. Nutrient-free media were based on DMEM without glucose (LM 001-56, Welgene) and 1% PS. Nutrient-full and -free media were used to for nutritional stress, including normal, starvation, and replenishment conditions.

### 4.3. In Vitro Nutritional Stress Protocol

The protocols from previous studies were modified [34,35]. GLUTag cell was seeded on plate with nutrient-full media. For starvation condition, GLUTag cells were cultured with nutrient-free media for 2 h. For replenishment, GLUTag cells were cultured with nutrient-full media for 2 h after 2 h nutrient starvation.

### 4.4. GLP-1 Secretion Assay

GLP-1 secretion was determined by ELISA using an active GLP-1 ELISA kit (EGLP-35K, Merck at Kenilworth, NJ, USA) following the manufacturer’s protocol. GLUTag were seeded in 12-well plates and cultured in nutrient-full media for 24 h. Media were replaced with nutrient-free media or nutrient-full media to generate starvation or replenishment conditions, respectively. A total of 100 nM LDN193189 or 20 ng/mL BMP4 were treated for 24 h or 2 h with sitagliptin prior to ELISA assay.

### 4.5. Mitochondria Stress Test Assay

Oxygen consumption rate (OCR) was determined by Seahorse XFp analyzer (Agilent at Santa Clara, CA, USA) using Mitochondria Stress test kit (MST; 103010-100, Agilent) following the manufacturer’s protocol. A total of 2 μM oligomycin, 0.75 µM carbonyl cyanide-p-trifluoromethoxyphenylhydrazone (FCCP), and 1 μM rotenone/antimycin A were injected, mixed for 3 min, left for 3 min, measured for 3 min, and cycled three times.

### 4.6. Western Blotting

Protein was extracted from GLUTag using EBC200 lysis buffer (200 mM NaCl, pH 8.0; 50 mM Tris-HCl; NP-40 0.4%) with protease inhibitor, 0.5 M EDTA (P3100, GenDEPOT at Hanam-si, Korea), and phosphatase inhibitor (P3200, GenDEPOT). Protein lysates were resolved on 10% or 15% SDS-polyacrylamide gels with protein standard (1610377, BIO-RAD at Hercules, CA, USA) and transferred to methanol-activated polyvinylidene difluoride membrane (1620177, BIO-RAD). Membrane blocked with 5% skim milk for 1h was incubated with diluted antibodies overnight at 4 °C. The anti-ID1 (sc-133104, Santa Cruz Biotechnology at Dallas, TX, USA), anti-ID4 (sc-365656, Santa Cruz Biotechnology), anti-SMAD1/5/9 (PA1-41238, Invitrogen), and anti-p-SMAD1/5/9 (13820T, Cell Signaling Technology at Danvers, MA, USA) were diluted to 1:1000, and anti-β-Actin (Santa Cruz Biotechnology) was diluted to 1:5000. Membrane was washed using Tris-buffered saline with 0.1% Tween 20 (TBST), and incubated with horseradish peroxidase-linked horse anti-mouse antibody (1:10,000 diluted; 7076S, Cell Signaling Technology) or horseradish peroxidase-linked goat anti-rabbit antibody (1:10,000 diluted; ab6721, Abcam at Cambridge, UK) for 1h. Membrane was washed five times per 10 m using TBST and detected by Clarity Max Western ECL Substrate (1705062, BIO-RAD). Quantification of immunoblot band intensity was analyzed by ImageJ (version 1.8.0_172; National Institute of Health).

### 4.7. Reverse Transcription PCR and Quantitative PCR

Total RNA was isolated from GLUTag by TRIZol (15596-018, Invitrogen at Carlsbad, CA, USA), following the manufacturer’s protocol. cDNA was synthesized from 1 μg of total RNA using ImProm-II Reverse Transcriptase (A3803, Promega at Madison, WI, USA). Quantitative PCR (qPCR) was performed using TOPreal qPCR 2X PreMIX (SYBR Green; RT501M, Enzynomics at Daejeon, Korea) with each primer pair following 40 cycles of denaturation 95 °C for 10 s, annealing at 58 °C for 15 s, and extension at 72 °C for 30 s. The result was normalized to 36b4 mRNA expression level. The primers were 5′-CGTCCTCGTTGGAGTGACA-3′ (36B4 sense), 5′-CGGTGCGTCAGGGATTG-3′ (36B4 antisense), 5′- CCTAGCTGTTCGCTGAAGGC-3′ (ID1 sense), 5′-CTCCGACAGACCAAGTACCAC-3′ (ID1 antisense), 5′-CAGTGCGATATGAACGACTGC-3′ (ID4 sense), 5′-GACTTTCTTGTTGGGCGGGAT-3′ (ID4 antisense) 5′-GCCAGCACTATCTACACATCC-3′ (NOG sense), and 5′-GCCAGCACTATCTACACATCC-3′ (NOG antisense).

### 4.8. RNA Sequencing

Total RNA samples were duplicated and transported to Macrogen Inc. (Seoul, Korea; www.macrogen.com). First, a library was constructed using TruSeq Stranded mRNA LT Sample Prep Kit (Illumina at San Diego, CA, USA) following TruSeq Stranded mRNA Sample Preparation Guide, part #15031047 Rev. E. Second, sequencing was performed by NovaSeq6000 (Illumina) using a NovaSeq 6000 S4 Reagent Kit (Illumina) following NovaSeq 6000 System User Guide Document #1000000019358 v02 protocol. Then the sequence was qualified by FastQC v0.11.7, trimmed by Trimmomatic v0.38, and mapped by HISAT2. It assembled gene and transcript expression level to read count or fragments per kilobase of transcript per million mapped reads (FPKM) by StringTie v1.3.4d. Trimmed mean of M-value (TMM) normalization was performed to reduce systematic bias using read count by edgeR package library. Finally, differentially expressed genes (DEG) were estimated by edgeR.

### 4.9. Visualization and Analyzation

Visualization and analyzation were based on DEG sorted by fold change (FC; |FC| ≥ 2) and *p*-value (*p*; *p* ≤ 0.05). DAVID algorithm (https://david.ncifcrf.gov/summary.jsp) was performed for KEGG (http://www.genome.jp/kegg) enrichment analysis. Gene set enrichment analysis (GSEA) was performed by GSEA Preranked tool by Macrogen Inc. Protein–protein interaction (PPI) was based on the String database (https://string-db.org/) and visualized using Cytoscape (https://cytoscape.org/). PPI was sorted by combined score (combined score ≥ 0.4 were considered as the threshold value) and connecting total genes counts (>3 counts). Volcano plots were visualized by log2(FC) and −log10(*p*-value). Plots were marked as 10 if −log10(*p*-value) was greater than 10.

### 4.10. siRNA

GLUTag cells were seeded on 12-well plates with growth media for 24 h. Then, 20 nM siRNAs against ID1 and ID4, and non-specific siRNA (Dharmacon at Lafayette, CO, USA) were transfected with lipofectamine RNAi MAX (13778075, Invitrogen) following manufacturer’s protocol for 48 h.

### 4.11. Statistical Analyses

Statistical analysis was performed by a one-way ANOVA test when there were more than two groups, and by Student’s *t*-test when two groups were analyzed. The *p*-values below 0.05 were marked as statistically significant (* *p* < 0.05, ** *p* < 0.01, *** *p* < 0.001). All values were visualized by means with standard error of the mean (SEM).

## 5. Conclusions

In this paper, we showed that nutritional stress rapidly modulates GLP-1 secretion in GLUTag cells. Interestingly, we also demonstrated that ID1-mediated BMP signaling pathway is crucial to control GLP-1 secretion in response to nutrient status in GLUTag cells. Inhibition of BMP signaling pathway using LDN193189 remarkably reduced GLP-1 secretion, whereas activation of BMP signaling pathway with BMP4 markedly potentiated GLP-1 secretion in response to nutritional replenishment. Altogether, we demonstrated that BMP signaling pathway is a novel molecular mechanism to control GLP-1 secretion in L cells. Development of selective agonist to potentiate BMP signaling pathway in L cells would be a novel therapeutic strategy to potentiate GLP-1 secretion, leading to restored glycemic homeostasis in T2D patients.

## Figures and Tables

**Figure 1 ijms-21-03824-f001:**
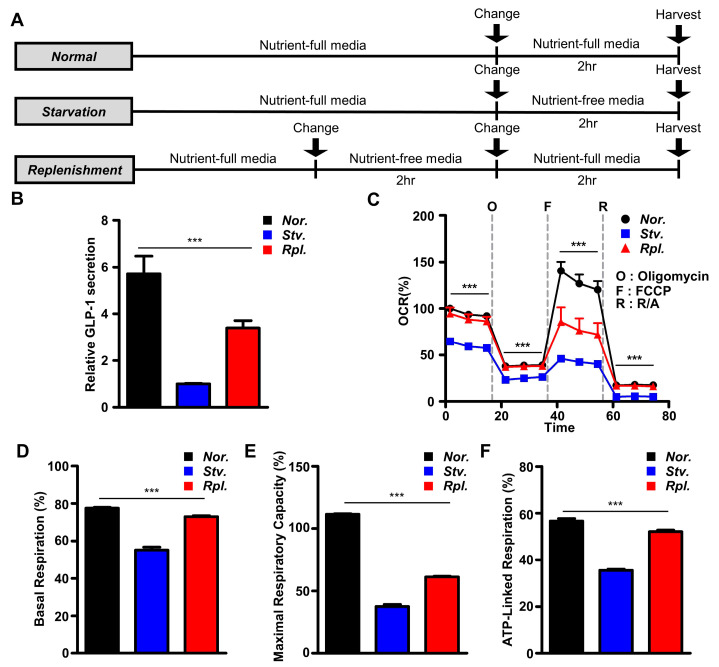
Nutritional stress rapidly controls glucagon-like peptide-1 (GLP-1) secretion and cellular bioenergetics in GLUTag cells. (**A**) Experimental scheme for in vitro studies. (**B**) Level of GLP-1 secretion in response to different nutritional status. All data were analyzed by GLP-1 ELISA (*n* = 9). (**C**) Cell respiration of oxygen consumption rate (OCR) in vitro (*n* = 3–6). (**D**) Basal respiration (*n* = 3–6). (**E**) Maximal respiratory capacity (*n* = 3–6). (**F**) ATP-linked respiration calculated from OCR (*n* = 3–6). Data represent the mean ± SEM. *** *p* < 0.001; one-way ANOVA.

**Figure 2 ijms-21-03824-f002:**
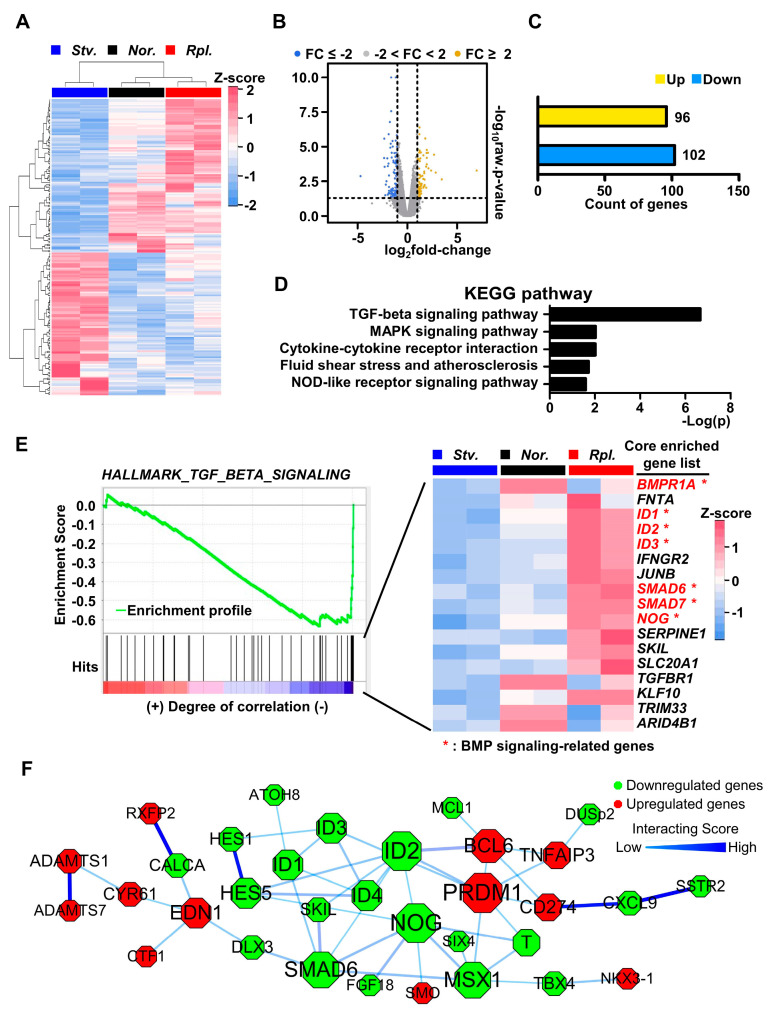
Nutrient starvation represses bone morphogenetic protein (BMP) signaling pathway in GLUTag cells. (**A**) Clustering of transcriptome analysis of starvation differentially expressed genes (DEG) versus normal and replenishment DEG. (**B**) Volcano plot of starvation DEG versus normal and replenishment DEG (**C**) Up- and down-regulated DEG of starvation normalized by DEGs of normal and replenishment. (**D**) KEGG pathway of DEG of starvation. (**E**) Gene set enrichment analysis with heatmap of core enriched gene expression profiles. (**F**) Functional protein association network analysis using STRING.

**Figure 3 ijms-21-03824-f003:**
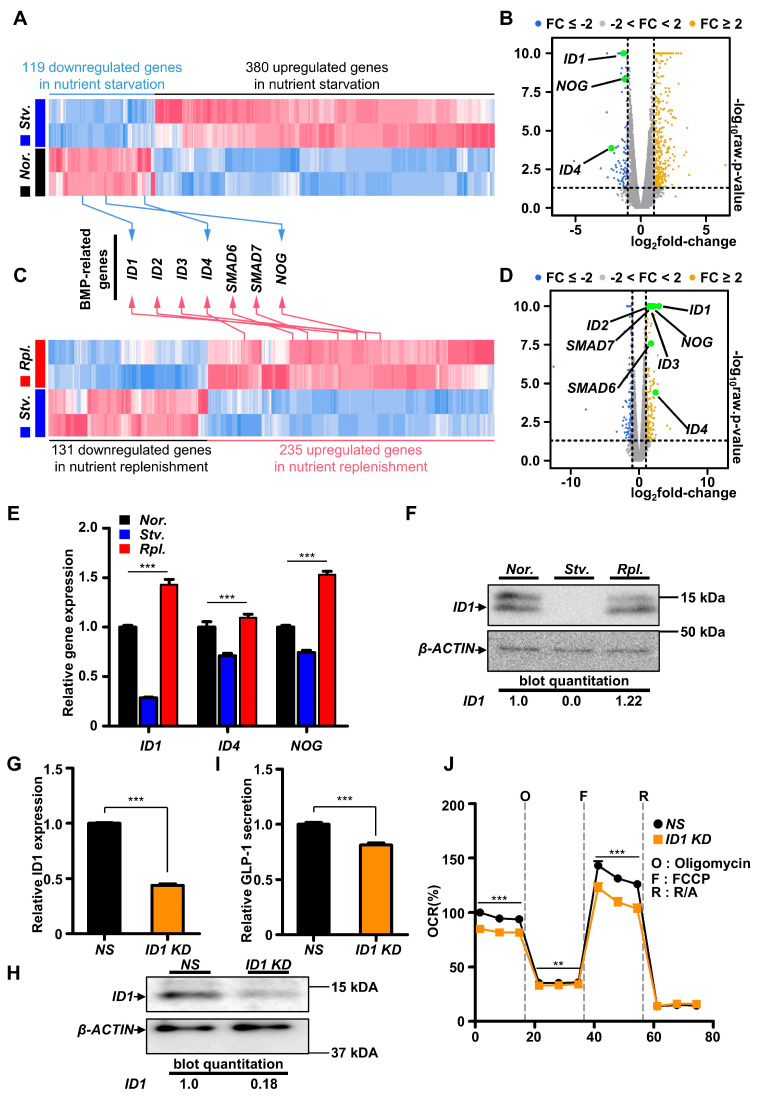
Inhibitor of DNA binding (ID)1-mediated BMP signaling modulated GLP-1 secretion in GLUTag cells. (**A**) Heatmap for normal versus starvation gene expression profile with overlapped BMP-related hub genes from protein association network. (**B**) Volcano plot of normal versus starvation gene expression profiles. (**C**) Heatmap for starvation versus replenishment gene expression profile with overlapped BMP-related hub genes from protein association network. (**D**) Volcano plot of starvation versus replenishment gene expression profiles. (**E**) mRNA expression level of ID1, ID4, and NOG according to nutrient status (*n* = 4−7). (**F**) ID1 protein level according to nutrient status. (**G**) mRNA level of ID1 gene with siRNA (*n* = 4) (NS; non-specific siRNA; KD; knockdown with targeted siRNA). (**H**) Protein level of ID1 with siRNA. (**I**) Basal GLP-1 secretion in GLUTag cells with ID1 siRNA (*n* = 12). (**J**) Cellular bioenergetics of GLUTag cells with ID1 siRNA (*n* = 6). Data represent the mean ± SEM. ** *p* < 0.01; *** *p* < 0.001; one-way ANOVA was performed for (E) and (J). Student’s *t*-test was performed for (G) and (I).

**Figure 4 ijms-21-03824-f004:**
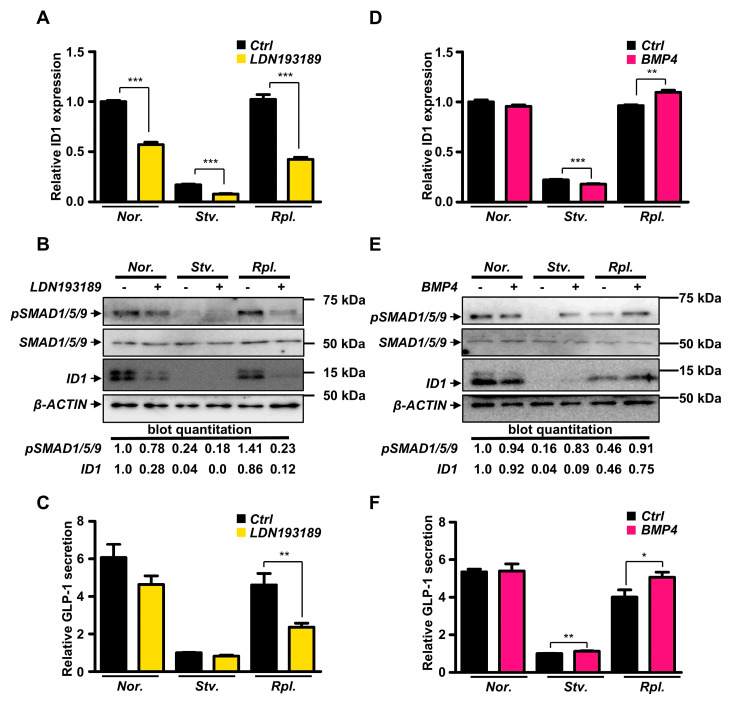
Modulation of BMP signaling pathway controlled GLP-1 secretion in GLUTag cells. (**A**–**C**) GLUTag cells were pre-treated with 10 nM LDN193189 for 24 h. (**A**) mRNA expression of ID1 gene with LDN193189 (*n* = 4). (**B**) Protein level of ID1 gene with LDN193189. (**C**) GLP-1 secretion with LDN193189 (*n* = 9). (**D**−**F**) GLUTag cells were pre-treated with 20 ng/mL of BMP4 for 2 h. (**D**) mRNA expression of ID1 gene with BMP4 (*n* = 3). (**E**) Protein level of ID1 gene with BMP4. (**F**) GLP-1 secretion with BMP4 (*n* = 6). Data represent the mean ± SEM. * *p* < 0.05; ** *p* < 0.01; *** *p* < 0.001; Student’s *t*-test.

**Figure 5 ijms-21-03824-f005:**
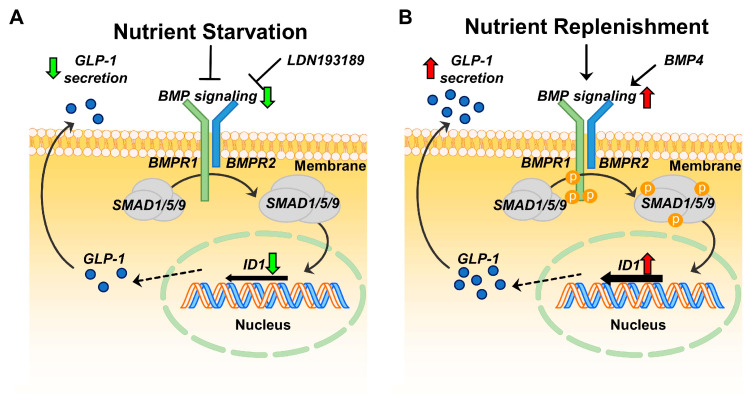
Molecular mechanisms of ID1-mediated BMP signaling pathway to control GLP-1 secretion in GLUTag cells. Change of nutrient status modulated ID1-mediated GLP-1 secretion in GLUTag cells. (**A**) Inhibition of BMP signaling by nutrient starvation and LDN193189 suppressed GLP-1 secretion. (**B**) Activation of BMP signaling by nutrient replenishment and BMP4 potentiated GLP-1 secretion in GLUTag cells.

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
