# Peer review of "ID1-Mediated BMP Signaling Pathway Potentiates Glucagon-Like Peptide-1 Secretion in Response to Nutrient Replenishment"

_ijms, 2020, doi:10.3390/ijms21113824_

Round 1

Reviewer 1 Report

In this paper, the authors examined the role of BMP signaling and ID1 protein in potentiating the secretion of GLP-1. The paper is well written and demonstrates that ID genes and BMP signaling pathways members are regulated under conditions of nutrient starvation and replenishment.

However, there are some concerns and questions that would improve the clarity of the paper.

i) In Figure 1A, the authors have a graphic showing the protocol for starvation and replenishment. How did the authors come up with the time frame for nutrient starvation and replenishment? Are there published papers that support the protocol this as a good model to mimic metabolic stress during obesity and T2D? It would be good to include the citation in the methods. 

2) In all the figures, although statistical analysis are shown, the N values (number of biological replicates) are not included. Were these experiments repeated on multiple biological replicates?How many technical replicates were included in each experiment? 

For example : The qPCR protocol mentions duplication of the total RNA - does that mean that the data shown (2 lanes of each data) come from one biological sample are are technical replicates or are they from two biological samples? Similarly, even though there are standard deviations included in each figure, it is unclear if the data for OCR comes from one experiment with multiple replicates or multiple experiments. 

3) On line 212 - 214, in describing the transcriptome data, the authors mention that the gene expression profiles of normal are very similar to those of nutrient replenished. Looking at the figure, it appears that although there is some similarity between normal and replenished -they do not appear to be similar. It would good for authors to use something like a Venn diagram to show the overlap between the genes in normal and replenished. This raises the question also about period of starvation - did they try increased starvation before replenishment - were the gene expression profiles similar to normal in those cases?

3) In the results section on line 266 - authors talk about the suggestion that ID1-mediated BMP signaling pathway was crucial. However, the figure referenced only shows that SMADs and ID genes are altered. The data show that ID1 KO decreases GLP1 but nothing here about it being downstream of BMP signaling. It would be good to edit that sentence. 

4) Since there is only a small effect on GLP-1 secretion with ID1 KO, did the authors examine other ID gene KO in their study - do they see an additive effect? A study by Sharma et al (Biochimie. 2015 May; 112: 139–150.) suggested that ID 4 could be an inhibitor for other IDs. It would good for the authors to address lack of greater inhibition of GLP-1 in their discussion.

5) Is the decrease in P-SMAD expression in Figure 4B reproducible and statistically significant in the presence of the inhibitor under normal conditions? Similarly, does Figure 4E show a change in ID1 protein expression upon BMP4 exposure? Including quantitation of the intensity on western blot and N values for the figure shown in Figure 4B and Figure 4D would help.

6) In Figure 4D, it appears from the figure that ID1 expression slightly decreases in the presence of BMP4 under starvation condition. However, there is a slight increase in GLP1 expression upon addition of BMP-4 under the same conditions, which does not agree with the previous suggestion that decrease in ID1 causes a decrease in GLP1. The authors do not address this in the paper. Is this change in ID1 and GLP1 upon addition of BMP4 under starvation condition reproducible? The authors need to address this part of the data in their results and reconcile it in the discussion.

7) Discussion can be enhanced by addressing some of the issues raised above.

8) Minor : In figure legend for Fig 4 - authors need to change (D) on line 310 to (E). 

Reviewer 2 Report

In submitted manuscript, authors described that GLP-1 secretions was increased by BMP pathway in nutrient replenishment condition. In particular, they showed that ID1-mediated BMP signalling pathway has a pivotal role in the control GLP-1 secretion.

The manuscript is well written and presented nicely; however, there are some gaps that the authors should try to fill to complete their study and improve the manuscript.

  • Authors indicated that ID1, ID4 and NOG genes were remarkably downregulated in nutrient starvation, while nutrient replenishment rescue these genes. After, authors showed only ID1 protein levels in the different conditions. Authors should perform analyse on ID4 and NOG protein levels. Moreover, has the inhibition of ID4 and NOG by siRNA got a similar effect of ID1 inhibition by siRNA?
  • Authors activated/inhibited BMP4 pathway and they analysed SMADs and ID1 protein levels. In my opinion, authors should analyse if ID4 and NOG have regulated by BMP signalling pathway.
  • Could ID1, ID4, and NOG have a synergic effect on GLP-1 secretions?
  • Authors should improve their discussion.
  • On page 11 lines 340-341 authors reported that cAMP signalling pathway was stimulated by bile acid-activated TGR5 in order to potentiate GLP-1 secretion. Has somebody never described about modification on GLP-1 secretion in biliary, hepatic or pancreatic disease showing bile acid altered compositions?

Round 2

Reviewer 2 Report

Authors answered in good way in this new manuscript form. I would like to suggest to accept the manuscript in this new form.